# On the Stark Broadening of Be II Spectral Lines

**Milan S. Dimitrijević [1,2], Magdalena Christova [3,*] and Sylvie Sahal-Bréchot [2]**

[1]    Astronomical Observatory, Volgina 7, 11060 Belgrade, Serbia; mdimitrijevic@aob.rs
[2]    Sorbonne Université, Observatoire de Paris, Université PSL, CNRS, LERMA, F-92190 Meudon, France; sylvie.sahal-brechot@obspm.fr
[3]    Department of Applied Physics, Technical University–Sofia, 1000 Sofia, Bulgaria
[*]    Correspondence: mchristo@tu-sofia.bg

**Abstract:** Calculated Stark broadening parameters of singly ionized beryllium spectral lines have been reported. Three spectral series have been studied within semiclassical perturbation theory. The plasma conditions cover temperatures from 2500 to 50,000 K and perturber densities $10^{11}$ cm$^{-3}$ and $10^{13}$ cm$^{-3}$. The influence of the temperature and the role of the perturbers (electrons, protons and He$^+$ ions) on the Stark width and shift have been discussed. Results could be useful for plasma diagnostics in astrophysics, laboratory, and industrial plasmas.

**Dataset:** Supplementary File.

**Keywords:** atomic data; atomic processes; line formation

## 1. Introduction

Atomic and spectroscopic data of light elements are of great importance in astrophysics and cosmology. LiBeB abundance is closely related to major questions concerning primordial nucleogenesis, stellar structure, mixing between atmosphere and interior, evolution, etc. [1–5]. It is well known that the abundance of the chemical elements versus the mass number is a notably decreasing curve [6,7]. The LiBeB trio is exceptional from this general trend in nature, because these elements are simple and rare. The mystery of lower light element abundance remains unresolved up to today [3]. What is the origin of these elements?; Are they generated in the normal course of stellar nucleosynthesis?; How they are destroyed?; these are all questions still unanswered. The three fragile light nuclei burn in the same $(p, \alpha)$ process and undergo nuclear reactions at relatively low temperatures, estimated at near 2.5, 3.5, and $5 \times 10^6$ K for densities similar to those in the Sun. In solar-type stars, these temperatures are reached not far below the convection zone and well outside the core, and circulation and destruction of the light elements can result in observable abundance changes. Observations of these changes can provide an invaluable probe of stellar structure and mixing. Be and B have been observed in extreme Population II with low Z, in a number of low metallicity halo dwarf stars [5]. Some papers suggest that observed Be and B were generated by cosmic-ray spallation in the early Galaxy, and the standard model of primordial nucleosynthesis is unable to produce significant yields of both light elements [5]. According to [3], there are no theoretical explanations for the reduction in the abundances, a trend of decreasing with effective temperature and a dip at Teff ~6600 K in F, G, and K-dwarfs that have been found in the Hyades and other old clusters. Interactions between emitting atoms and surrounding electrons and ions result in Stark broadening of spectral lines. This broadening mechanism of line profiles is usually a principle one in the case of white dwarfs, and is of interest for main sequence stars from A type and late B type, and sometimes dominant ones [8,9]. Stark broadening parameters of

Be II lines could serve effectively for the adequate modelling of stellar objects, opacity calculations, and diagnostics of astrophysical objects, laboratory and technological plasmas.

Previous Stark broadening calculations based on the semiclassical perturbation theory [10,11] of Be II lines have been performed in [12,13]. Here, an additional data set of Stark broadening parameters (widths and shifts) for Be II transitions, not included in [12,13], has been calculated. This dataset is used to compare the importance of the broadening due to electron-, proton-, and ionized helium-impacts, in function of temperature and to investigate the behavior of Stark broadening parameters within three spectral series.

## 2. Data Description and Method of Research

A dataset with new results for Stark broadening parameters of Be II spectral lines has been provided (Table S1). The calculations were performed using the semiclassical perturbation theory [10,11,14].

Within this theory, in the case of a charged emitter/absorber, full width at half intensity maximum (FWHM) $W$, and shift $d$ of an isolated spectral line originating from the transition between the initial level $i$ and the final level $f$ are given as:

$$W = 2N \int_0^\infty vf(v)dv \left( \sum_{i' \neq i} \sigma_{ii'}(v) + \sum_{f' \neq f} \sigma_{ff'}(v) + \sigma_{el} \right)$$
$$d = N \int_0^\infty vf(v)dv \int_{R_3}^{R_d} 2\pi\rho d\rho \sin(2\varphi_p) \tag{1}$$

where with $i'$ and $f'$ are denoted perturbing levels, $N$, $v$, and $f(v)$ are the perturber density, velocity, and the Maxwellian distribution of perturber velocities, respectively, and $\rho$ is the impact parameter of the perturber colliding with the emitter/absorber. The inelastic cross sections $\sigma_{kk'}(v)$, $k = i,f$, can be expressed by an integration of the transition probability $P_{kk'}(\rho,v)$:

$$\sum_{i' \neq i} \sigma_{ii'}(v) = \frac{1}{2}\pi R_1^2 + \int_{R_1}^{R_d} 2\pi\rho d\rho \sum_{i' \neq i} P_{ii'}(\rho, v) \tag{2}$$

and the elastic contribution to the width is:

$$\sigma_{el} = 2\pi R_2^2 + \int_{R_2}^{R_d} 2\pi\rho d\rho \left( \sin^2\delta \right) + \sigma_r \tag{3}$$

$$\delta = \left( \varphi_p^2 + \varphi_q^2 \right)^{1/2} \tag{4}$$

In the above Equations, $\sigma_{el}$ is the elastic cross section, while $\varphi_p$ and $\varphi_q$ are phase shifts due to the polarization and quadrupolar potential (see Section 3 of Chapter 2 in [10]). For the cut-offs $R_1$, $R_2$, and $R_D$, see Section 1 of Chapter 3 in [11]. The quantity $\sigma_r$ denotes the contribution of Feshbach resonances (see [14] and references therein), which concerns only electron-impact widths.

For isolated lines, the profile $F(\omega)$ has Lorentzian form:

$$F(\omega) = \frac{W/(2\pi)}{\left(\omega - \omega_{if} - d\right)^2 + (W/2)^2} \tag{5}$$

where $\omega_{if} = E_i - E_f/\hbar$ and $E_i$, $E_f$ are the energies of the initial and final state. Therefore, if we know the Stark broadening parameters, width $W$ and shift $d$, we can determine the spectral line profile.

Here, we calculated Stark broadening parameters for three spectral series: $1s^2 4d$–$1s^2 np$; $1s^2 4d$–$1s^2 nf$ and $1s^2 4f$–$1s^2 nd$, where $n = 6$–$8$. The temperature varied from 2500 K to 50,000 K and the perturber density was $10^{11}$ cm$^{-3}$ and $10^{13}$ cm$^{-3}$. The values of energy levels were taken from [15], and the oscillator strengths were calculated using [16]. For atoms such as beryllium an error around 20% was expected [17]. The dataset (Table S1) contains full Stark widths at half intensity

maximum and shifts of Be II spectral lines due to collisions with electrons, protons and ionized helium, the main constituents of stellar atmospheres.

## 3. Results and Discussion

### 3.1. Series $1s^2 4d$–$1s^2 np$

#### 3.1.1. Temperature Dependence

Previous calculations [12,13] of Stark broadening parameters for the same series include results for one transition: $1s^2 4d$–$1s^2 5p$. In order to complete the data, using available energy values from [15], results for transitions from higher levels have been obtained. With the increase in the principal quantum number of the upper atomic energy level within a spectral series, the maximal perturber density for which the impact approximation is valid, decreases. For a density of $10^{13}$ cm$^{-3}$, it is valid for all electron-impact broadening parameters within the considered data set, but not for all other widths and shifts in the case of collisions with heavier particles, protons and helium ions. Therefore, we performed calculations for $10^{11}$ cm$^{-3}$ too, where impact approximation was valid for all perturbers within the considered data set. Moreover, these electron densities are typical for stellar atmospheres and for lower ones the linear extrapolation could be applied. In Figure 1a,b the temperature ($T$) dependence of Stark width and shift due to collisions with electrons, protons and ionized helium for $1s^2 4d$–$1s^2 6p$ transition have been illustrated.

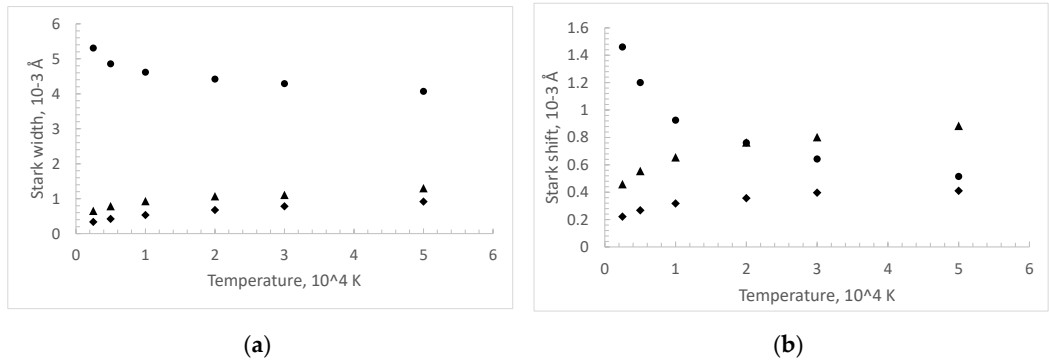

(a)                                                                 (b)

**Figure 1.** Stark broadening width (**a**) and shift (**b**) for multiplet $1s^2 4d^2$D–$1s^2 6p^2$P° (6638.3 Å) versus temperature from different types of perturbers: electrons—circle; protons—rhombus; ionized helium ions—triangle. Perturber density was $1 \times 10^{13}$ cm$^{-3}$.

The dominant width was from the impacts with electrons. It decreased very slowly with the temperature. The broadening due to interactions with protons and He$^+$ ions were almost the same and they had the same trend, very slow increases with $T$. These values were notably lower (within an order) than the electron impact width. We observed the same trends with temperature for impact shifts, but the values for the three perturbers were much closer except for temperatures below 10,000 K, where the electron-impact shift started to dominate. For 20,000 K, the shifts due to electron- and He$^+$ ion-impacts were the same. For higher temperatures, the He$^+$ shifts were highest.

The Stark broadening parameters for $1s^2 4d$–$1s^2 8p$ transition are depicted in Figure 2a,b.

The trends of widths due to different perturbers with temperature are the same as in the previous case. The values converged with increasing temperature, particularly electron width and He$^+$ width. Comparing the shifts for the two transitions, we observed analogy in the behavior and differences in the values for different components. The impact component from He$^+$ ions was dominant for practically all temperatures. For temperatures above 15,000 K, proton shift also overcame the electron shift.

If we compare Stark broadening parameters for $n = 6$ and $n = 8$, we can see that the influence of collisions with protons and helium ions, compared to the collisions with electrons, increases for higher $n$, so that the shift becomes dominant with the increase in temperature.

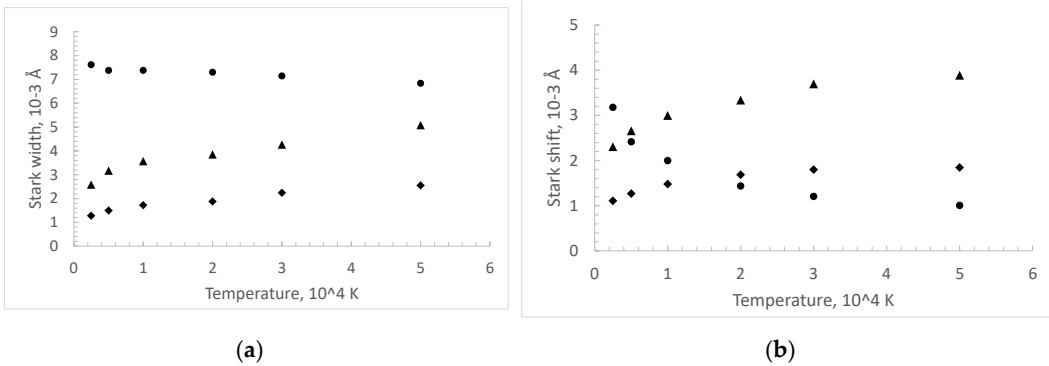

(**a**)    (**b**)

**Figure 2.** Stark broadening width (**a**) and shift (**b**) for multiplet 1s24d2D–1s28p2Po (4874.1 Å) versus temperature from different types of perturbers: electrons—circle; protons—rhombus; ionized helium ions—triangle. Perturber density was $1 \times 1013$ cm$^{-3}$.

### 3.1.2. Dependence on Principal Quantum Number

In Figure 3a,b dependence of Stark broadening parameters on the principal quantum number of the upper atomic energy level of the spectral series $1s^2 4d^2D–1s^2 np^2P^o$, for $n = 5–8$ have been presented for electron-impact broadening. Stark broadening parameters, widths, and shifts have been expressed in angular frequency units and as decimal logarithms. The values for $n = 5$, have been extrapolated linearly to the electron density of $10^{11}$ cm$^{-3}$. Namely, from references [12,13], where Stark broadening parameters for perturber densities from $10^{13}$ cm$^{-3}$ to $10^{19}$ cm$^{-3}$ are provided, it follows that the dependence on the electron density is linear towards the lower densities. Their increasing with principal quantum number is very regular.

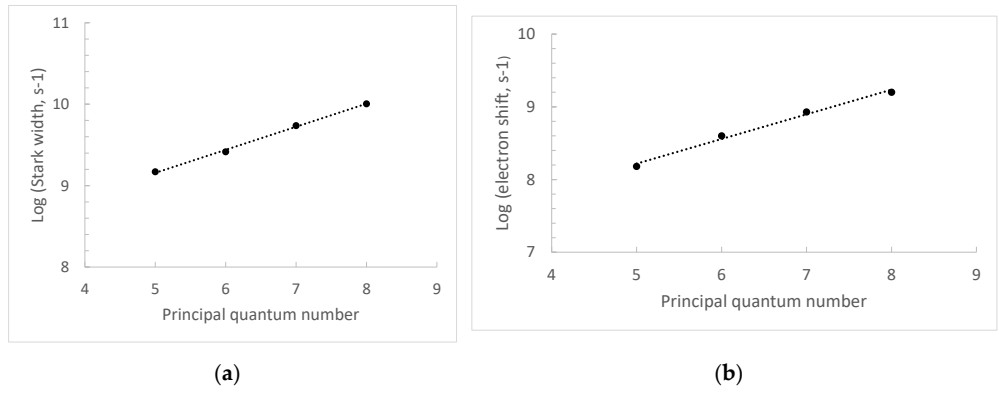

(**a**)    (**b**)

**Figure 3.** Decimal logarithm of full Stark width (**a**) and electron-impact shift (**b**) for spectral lines within the $2s^2 4d–2s^2 np$ ($n = 5–8$) spectral series versus principal quantum number. Electron density was $1 \times 10^{11}$ cm$^{-3}$ and temperature was $1 \times 10^4$ K. The Stark full width at half intensity maximum (FWHM) values for $n = 5$ have been taken from [13].

In the case of proton-, and helium ion-impacts, the behavior with the increase in $n$ was the same, while for the shift, we had the same behavior for $n = 6–8$. For $n = 5$ the proton-impact shift was negative ($-0.101$ Å [13]), as was the helium ion-impact shift ($-0.0865$ Å [13]).

We confirmed that for all three spectral series the behavior of widths and shifts with the increase in the principal quantum number was regular, and for all three considered series the result was the same and the behavior was very similar. Therefore, we can conclude that for all three Be II spectral series, this regular behavior can be used for interpolation and extrapolation to estimate the missing values within a considered series, and for confirmation of experimental and theoretical results: for all considered perturbations concerning Stark widths, and for electron-impact shifts.

## 3.2. Series $1s^2 4d$–$1s^2 nf$

　　Comparison of electron impact widths and shifts versus temperature for three transitions is illustrated in Figure 4a,b. The behavior of electron widths was the same: they decreased notably for lower temperatures up to 20,000 K and slowly for higher *T*. In accordance with the theory, the greater values corresponded to the higher transition. It is visible that the width's gradient slightly increased with principal quantum number for lower temperatures. All shifts were positive, red shifts. Figure 4b shows that shift trends were the same. Between temperatures 2500 K and 5000 K there was a big jump followed by decrease towards the higher temperatures. For temperatures above 10,000 K, electron shifts had practically the same values. As a difference from the line width, where contributions of virtual transitions to all energy levels were positive, in the case of shift we had a sum of positive and negative contributions, so that the behavior with temperature could be more complicated. In the considered series, the closest perturbing level to *nf* levels was *nd* level, with negative contribution to the shift. With the increase in temperature the role of a particular level decreases, therefore we first had an increase due to a relative decrease in negative contribution with an increasing role of levels contributing positively, and then the characteristic decreased with the increase in temperature.

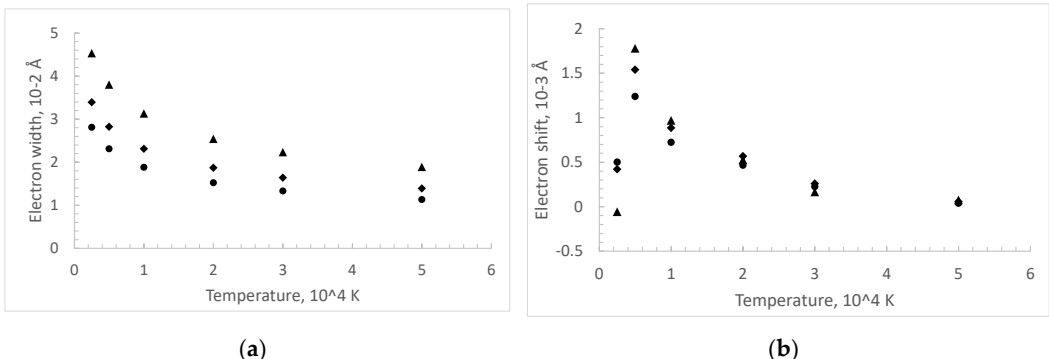

(**a**)　　　　　　　　　　　　　　　　　　　　　　　　　(**b**)

**Figure 4.** Stark broadening parameters for electron-impacts: width (**a**) and shift (**b**) versus temperature for the $2s^2 4d$–$2s^2 nf$ series: *n* = 6—circle; *n* = 7—rhombus; *n* = 8—triangle. Perturber density was $1 \times 10^{13}$ cm$^{-3}$.

## 3.3. Series $1s^2 4f$–$1s^2 nd$

　　The temperature dependence of Stark broadening parameters for corresponding transitions is demonstrated in Figure 5a,b, respectively.

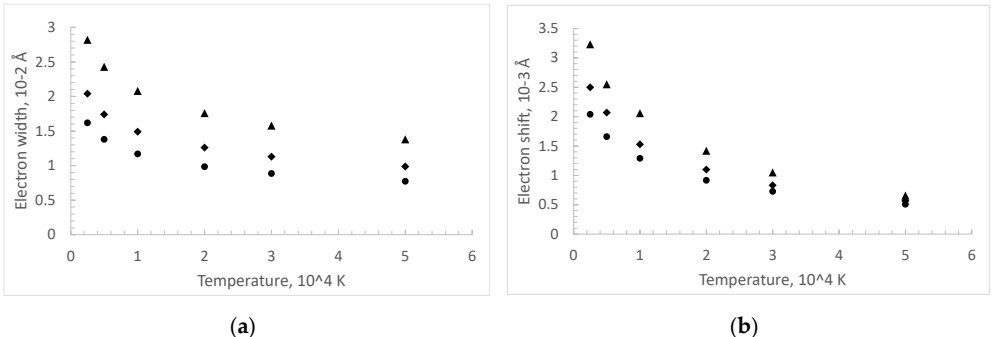

(**a**)　　　　　　　　　　　　　　　　　　　　　　　　　(**b**)

**Figure 5.** Electron-impact width (**a**) and shift (**b**) versus temperature for the $2s^2 4f$–$2s^2 nd$ series: *n* = 6—circle; *n* = 7—rhombus; *n* = 8—triangle. Electron density is $1 \times 10^{13}$ cm$^{-3}$.

　　Both width and shift decreased with *T*. The width's decreasing was uniform for all three transitions, while shifts decreased and converged to practically the same value for 50,000 K.

If we compare the behavior of Stark broadening parameters within the spectral series $1s^2 4d$–$1s^2 nf$ and $1s^2 4f$–$1s^2 nd$ we can see that the behavior of widths due to collisions with electrons is very similar and that for all considered temperatures they increased with the increasing of the principal quantum number in a similar manner, and decreased uniformly with the increase in temperature. On the other hand, the shifts within these two series had different behavior, especially at a low temperature limit. However, in the case of the $1s^2 4f$–$1s^2 nd$ series, shifts uniformly decreased with the increase in temperature; in the $1s^2 4d$–$1s^2 nf$ series they considerably increased from $T = 2500$ K to $T = 5000$ K, and then uniformly decreased. A common characteristic for both series was a shift convergence with temperature. The width values did not converge.

## 4. Conclusions

Stark broadening parameters, FWHM and shifts due to impacts with electrons, protons and helium ions have been obtained for three spectral series in the Be II spectrum, by using semiclassical perturbation theory [10,11,14]. The calculations were performed for a temperature interval from 2500 to 50,000 K, and electron densities of $10^{11}$ and $10^{13}$ cm$^{-3}$. The dependence of Stark broadening parameters with temperature and the role of different perturbers (electrons, protons and He$^+$ ions) on the Stark width and shift have been discussed. Additionally, the regularity of behavior of Stark broadening parameters within the three considered spectral series was confirmed, and it was found that such regularities can be used for the interpolation and extrapolation of missing values and for a confirmation of experimental and theoretical results.

The obtained results may be of interest in astrophysics, especially for investigations of stellar atmospheres, and in particular for the problem of LiBeB abundances, closely related to major questions concerning primordial nucleogenesis, stellar structure, mixing between atmosphere and interior, as well as the stellar evolution, but also for stellar opacities, radiative transfer, stellar atmospheres modelling and analysis, and synthesis of stellar spectra. These data may be also useful for laboratory plasma diagnostics, modelling and investigation, and for the examination of regularities and systematic trends of Stark broadening parameters.

**Supplementary Materials:** The following are available online at http://www.mdpi.com/2306-5729/5/4/106/s1, Table S1: Full Stark widths at half intensity maximum and shifts due to collisions with electrons, protons and ionized helium for perturber densities of $10^{11}$ cm$^{-3}$ and $10^{13}$ cm$^{-3}$.

**Author Contributions:** Conceptualization, M.S.D. and M.C.; software, S.S.-B.; validation, M.S.D. and S.S.-B.; formal analysis, M.C. and M.S.D.; writing—original draft preparation, M.C. and M.S.D.; writing—review and editing, M.S.D. and M.C.; supervision, M.S.D. and S.S.-B. All authors have read and agreed to the published version of the manuscript.

**Funding:** This research received no external funding.

**Acknowledgments:** Thanks to the Scientific and Research Sector of TU-Sofia for partial support.

**Conflicts of Interest:** The authors declare no conflict of interest.

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
