# Peer review of "On the Stark Broadening of Be II Spectral Lines"

_data_

Round 1
Reviewer 1 Report
I think that the paper (data descriptor) is worth to be published with minor amendments. Specifically I would suggest:
-'electron density' ( text lines 12,40,57),should be changed in 'perturber density'
-Authors should specify that the 'perturber density' in the presented calculations (10^11 and 10^13 cm-3) are much lower than that assumed in ref 8 ( 10^15 cm-3), possibly with few words motivating the choice.
-Authors should 'explain' how the point at n=5 in Fig.3 ( electron density 10^11 cm^-3 , is deduced from ref 8 ( perturber density 10^15 cm^-3)
Author Response
The answer is attached.

Reviewer 2 Report
In this work, the authors compute the Stark effect of Be II lines as a result of collisions of electrons, protons, and He ions. As a brief data report, the manuscript is basically fine but missing some information.
The scenario of such a computation is critical and missing. The authors mentioned their electron density, but no information on the densities of other species. Without an ionization degree, the electron density can be quite meaningless. What pressure and initial species composition is? The ionization rates and recombination rates can vary the electron and He ion densities dynamically and these rates are functions of collision cross sections which depends on the relative velocity of collisions. Therefore, EEDF as a critical information is also missing. If the collisions among electrons, He, and He ions are neglected, please add more statements to discuss these assumptions.
Other issues including the missing explanation of the sudden boost of shift at 0-10000K in Fig. 4(b). A detailed physical discussion should be added. Also, it will be better to add a brief description of the semiclassical computation method than citing the ref [5, 6, 9].
Finally, it will be better to add detailed applications to show the value of this work, rather than simply claim that “The obtained results may be of interest in astrophysics, especially for investigations of stellar atmospheres, but also for laboratory plasma diagnostics and research”. Adding some exact examples of using these results can significantly increase the impact of this work. The introduction and conclusion of the current version are very weak.
Considering these issues, I recommend a major revision of this manuscript.
Author Response
The answer is attached.

Round 2
Reviewer 2 Report
The authors have revised the manuscript properly according to my previous report. I recommend an acceptance for this version.